# Live imaging of *Yersinia* translocon formation and immune recognition in host cells

**Maren Rudolph**<sup></sup>, **Alexander Carsten**<sup></sup>, **Susanne Kulnik**, **Martin Aepfelbacher** [ID]<sup></sup>*, **Manuel Wolters** [ID]<sup></sup>*

Institute of Medical Microbiology, Virology and Hygiene, University Medical Center Hamburg Eppendorf, Hamburg, Germany

☉ These authors contributed equally to this work.

* m.aepfelbacher@uke.de (MA); m.wolters@uke.de (MW)

**Data Availability Statement:** All relevant data are within the manuscript and its Supporting Information files.

## Abstract

*Yersinia enterocolitica* employs a type three secretion system (T3SS) to translocate immunosuppressive effector proteins into host cells. To this end, the T3SS assembles a translocon/pore complex composed of the translocator proteins YopB and YopD in host cell membranes serving as an entry port for the effectors. The translocon is formed in a *Yersinia*-containing pre-phagosomal compartment that is connected to the extracellular space. As the phagosome matures, the translocon and the membrane damage it causes are recognized by the cell-autonomous immune system. We infected cells in the presence of fluorophore-labeled ALFA-tag-binding nanobodies with a *Y. enterocolitica* strain expressing YopD labeled with an ALFA-tag. Thereby we could record the integration of YopD into translocons and its intracellular fate in living host cells. YopD was integrated into translocons around 2 min after uptake of the bacteria into a phosphatidylinositol-4,5-bisphosphate enriched pre-phagosomal compartment and remained there for 27 min on average. Damaging of the phagosomal membrane as visualized with recruitment of GFP-tagged galectin-3 occurred in the mean around 14 min after translocon formation. Shortly after recruitment of galectin-3, guanylate-binding protein 1 (GBP-1) was recruited to phagosomes, which was accompanied by a decrease in the signal intensity of translocons, suggesting their degradation or disassembly. In sum, we were able for the first time to film the spatiotemporal dynamics of *Yersinia* T3SS translocon formation and degradation and its sensing by components of the cell-autonomous immune system.

## Author summary

Type 3 secretion systems (T3SS) are injection machines required for virulence of various bacteria. A translocon/pore complex is formed at the tip of the injection needle, which is required for translocation of effector proteins across the host cell membrane. To date, little information is available on the composition, structure, and regulation of the translocon. Here, we present a novel method that allows us to film the spatiotemporal dynamics of the *Yersinia* T3SS translocon by inserting a small peptide tag into the minor translocon protein YopD that can be bound with high affinity by a corresponding fluorophore-labeled

**Funding:** This study was supported by the Joachim Herz Foundation (www.joachim-herz-stiftung.de) given to AC. The funders had no role in study design, data collection and analysis, decision to publish, or preparation of the manuscript.

**Competing interests:** The authors have declared that no competing interests exist.

nanobody during the infection process. By this the formation and disassembly of the translocon and its sensing by components of the cell-autonomous immune system could be recorded with high temporal resolution. The described approach has the potential to provide insights into the T3SS of other bacterial species and is likely transferable also to other secretion systems.

## Introduction

Type three secretion systems (T3SSs) are multi-component, syringe-like nanomachines that enable the translocation of bacterial effector proteins across the bacterial envelope into eukaryotic host cells. Numerous human pathogenic bacteria such as *Yersinia*, *Pseudomonas*, *Chlamydia*, *Shigella* and *Salmonella* employ T3SS-mediated effector translocation to manipulate a variety of cellular processes, ultimately determining the nature of interaction with their hosts. T3SS effector proteins are diverse in structure and biochemical activities and vary considerably between species. In contrast, the T3SS machinery—also known as the injectisome—is highly conserved across different bacterial species and has been the subject of intensive structural and functional investigation [1–14].

Injectisomes can be separated into defined substructures such as the sorting platform, the export apparatus, the needle complex, the tip complex and the translocon (Fig 1A). The needle complex is a multi-ring cylindrical structure embedded in the bacterial envelope connected with a 30–70 nm long needle filament, forming a narrow channel, through which the translocator and effector proteins pass in an unfolded state [6,13]. The needle at its distal end transitions into the tip complex, which consists of several copies of a hydrophilic translocator protein (5 copies of LcrV in *Yersinia*) [9]. The tip complex is involved in host cell sensing and regulates the assembly of the translocon/pore complex [15].

The translocon of all investigated T3SSs consists of two hydrophobic translocator proteins, a major (YopB in *Yersinia*) and a minor translocator (YopD in *Yersinia*) harboring one and two transmembrane domains, respectively [16,17]. The two translocators are thought to form a heteromultimeric ring structure with an inner opening of approximately 2–4 nm in the host cell membrane [18–20]. Despite the central role of the translocon for effector translocation, many aspects of its regulation, assembly and composition have remained elusive. The hydrophobic nature of the translocators and the fact that the assembled translocon can only be studied when inserted into host cell membranes, up to now hindered its investigation due to a lack of suitable experimental approaches.

In a recent cryo-electron tomography study the host cell membrane embedded translocon of *Salmonella enterica* minicells was found to have a total diameter of 13.5 nm [11]. In our previous work translocons of *Yersinia enterocolitica* were imaged by super resolution immunofluorescence techniques (STED, SIM) using antibodies against the translocator proteins YopB and YopD. Thereby, the host cellular context that promotes translocon formation could be investigated, revealing that the translocons are formed upon uptake of the bacteria into a phosphatidylinositol-4,5-bisphosphate (PIP2)—enriched pre-phagosomal compartment/prevacuole, which is still connected to the extracellular space [21].

While these approaches provided a considerable degree of spatial resolution, none of them was suitable for time resolved imaging of translocons. Live imaging of bacterium-host cell interactions using fluorescence microscopy has become a key technology for understanding bacterial infection biology [22]. However, live imaging of translocon formation and processing in host cells has not yet been accomplished. This is likely due to the elaborate und highly

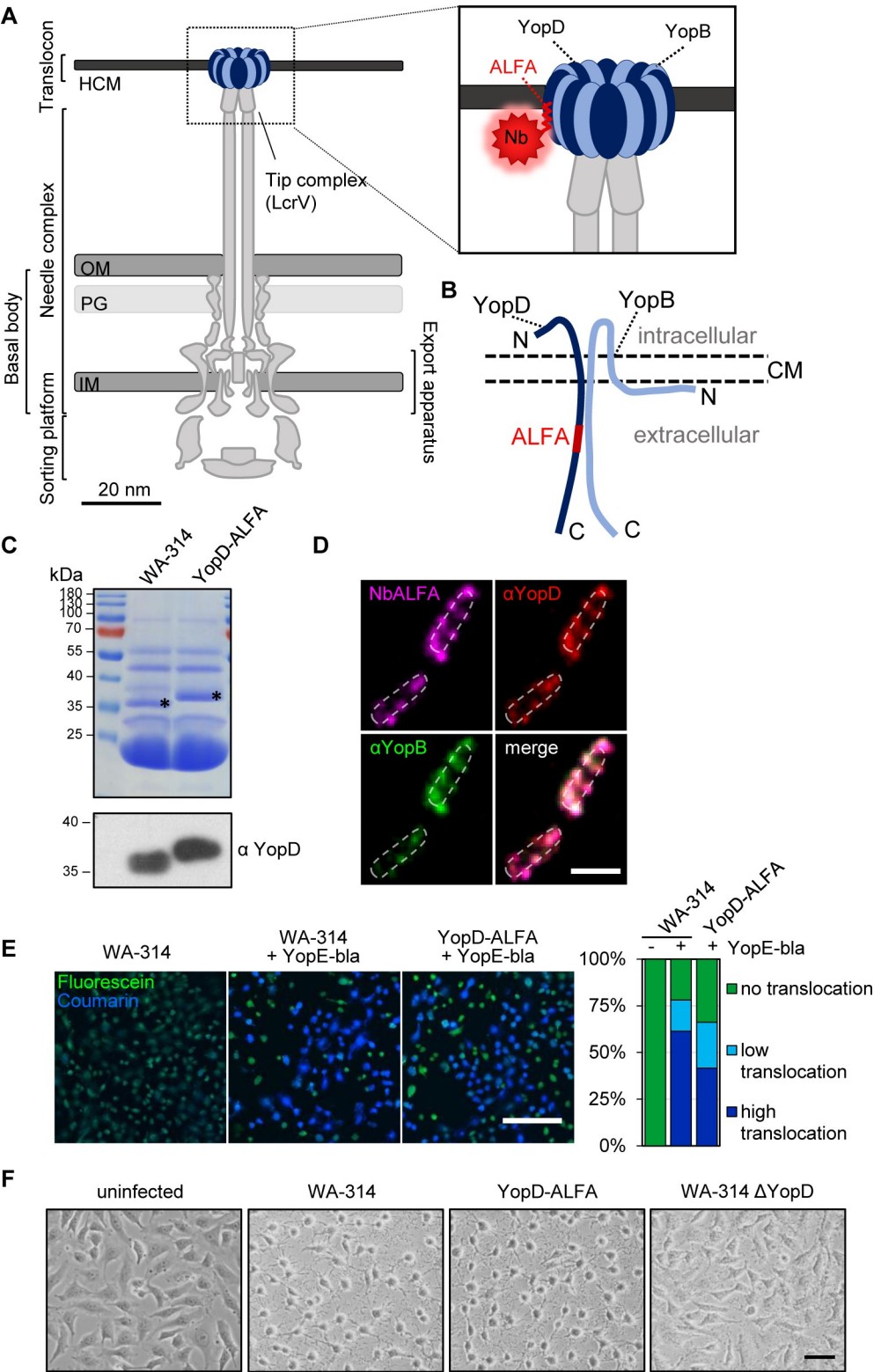

**Fig 1. Insertion of the ALFA-tag into YopD does not interfere with protein function and allows for nanobody-based staining of translocons. (A) Schematic representation of the T3SS in *Yersinia enterocolitica* with ALFA-tagged YopD.** The T3SS connecting the bacterial and host cell membranes. The enlargement shows the translocon with ALFA-tagged YopD labeled with a fluorescently tagged nanobody (NbALFA). IM: inner bacterial membrane. PG:

bacterial peptidoglycan layer. OM: outer bacterial membrane. HCM: host cell membrane. Adapted from [4,11]. **(B) Model of YopD-ALFA and YopB inserted into the host cell membrane.** The scheme is adapted from [23] and based on data on interactions of *Pseudomonas aeruginosa* PopD and PopB. The red box indicates the inserted ALFA-tag between amino acids 194 and 195 on the extracellular part of YopD. **(C) Released proteins of WA-314 and WA-314 YopD-ALFA.** Secreted proteins were precipitated from the culture supernatant and analyzed by Coomassie stained SDS gel (upper panel) and Western blot (lower panel) for their YopD content using specific antibodies. Black asterisks indicate the position of the YopD bands in the SDS gel. **(D) Staining of YopD-ALFA in translocons.** Rac1Q61L expressing HeLa cells were infected with WA-314 YopD-ALFA at an MOI of 10 for 1 h, fixed and host cell membranes were permeabilized with digitonin. Co-staining of translocon components was conducted with anti-YopB (shown in green) and anti-YopD (shown in red) antibodies and NbALFA-635 (shown in magenta). Scale bar: 2 μm. **(E) Comparison of effector protein translocation by β-lactamase assay.** HeLa cells pretreated with a cell permeant FRET dye (CCF4/AM) were infected for 1 h with WA-314, WA-314 pYopE-bla and WA-314 YopD-ALFA pYopE-bla at an MOI of 100 and imaged by confocal microscopy. Excitation of coumarin results in FRET to fluorescein in the uncleaved CCF4 emitting a green fluorescent signal. Cleavage of the cephalosporin core of CCF4 by the beta-lactamase tagged to a truncated YopE translocated into the host cell disrupts FRET and results in a blue fluorescent signal induced by the excitation of coumarin. Cells with incomplete CCF4 cleavage appear cyan. Scale bar: 200 μm. The percentage of green, cyan and blue cells was determined in one experiment from 354, 329 and 305 cells for WA-314, WA-314 pMK-bla and WA-314-YopD-ALFA pMK-bla, respectively. **(F) Cytotoxicity assay.** HeLa cells were infected for 1 h with WA-314, WA-314 YopD-ALFA and WA-314ΔYopD at an MOI of 100 and imaged by phase contrast microscopy. Depicted are phase contrast images of a representative experiment. Scale bar: 20 μm.

coordinated export of the hydrophobic translocator proteins through the T3SS needle and their interaction with the tip complex before they assemble a heteromultimeric translocon in the host cell membrane [16,23]. Thus, finding a label for translocon proteins that is e.g., suited for live cell imaging and super resolution and at the same time does not disturb translocon assembly has proven to be difficult. Fusion proteins of T3SS substrates with fluorescent proteins like GFP were shown to be resistant to T3SS-mediated unfolding and block the secretion path [12]. Several other tags (e.g. self-labeling enzymes Halo, CLIP, SNAP, split-GFP, 4Cys-tag/ FlAsH, iLOV) are secreted more effectively and have been used with varying degree of success for live imaging of translocated effectors [24–28].

We here report the first live cell imaging data of *Yersinia* translocon formation, immune sensing and processing by employing a *Yersinia* strain carrying a novel 13 amino acid peptide tag called ALFA-tag in the minor hydrophobic translocator YopD [29]. These data provide novel insights on the spatiotemporal dynamics and immune recognition of bacterial T3SS translocons.

## Results

### Characterization of *Y. enterocolitica* strain WA-314 YopD-ALFA

During infection with pathogenic yersiniae, a translocon/heteromultimeric pore complex composed of the translocator proteins YopB and YopD is integrated into host cell membranes, serving as an entry gate for the effector proteins (Fig 1A). In search of a method to visualize translocons in living cells using fluorescence microscopy, we inserted the ALFA-tag sequence (plus linker sequences) into the endogenous copy of *yopD* in the wild type strain WA-314 by CRISPR-Cas assisted recombineering, resulting in strain WA-314 YopD-ALFA (see Materials and Methods). The resulting YopD protein harbors the ALFA-tag between amino acids 194 and 195. The ALFA-tag insertion site is supposedly located in the extracellular part of YopD after it has integrated into the host cell membrane (Fig 1B) [23]. The 13 amino acid long ALFA-tag can be bound with high affinity by specific nanobodies (NbALFA) [29].

We first investigated whether WA-314 YopD-ALFA retains wild type functionalities by comparing its secretion-, translocon forming- and translocation capabilities as well as cytotoxic effect with the parental strain WA-314 (Fig 1C–1F). To analyze effector secretion, the low calcium response was exploited, which triggers secretion of translocator- and effector

proteins (*Yersinia* outer proteins; Yops) into the supernatant, when the bacteria are placed in calcium depleted medium at 37˚C [21–23]. SDS-PAGE and Western blot showed similar levels of total secreted proteins and secreted YopD-ALFA in WA-314 YopD-ALFA when compared to WA-314 (YopD-ALFA levels were compared to YopD levels), suggesting that protein secretion is unaffected in WA-314 YopD-ALFA (Fig 1C). Staining of WA-314 YopD-ALFA infected HeLa cells with fluorophore-labeled NbALFA revealed distinct fluorescence patches that were also detected by anti-YopD and anti-YopB antibodies (Fig 1D). Such patches have recently been shown to represent clusters of translocons [21]. Further, the ability of WA-314 YopD-ALFA to translocate effector proteins into host cells was examined using a well-established β-lactamase reporter system. This assay relies on cleavage of a cell permeant FRET dye (CCF4/AM) by a trans-located YopE β-lactamase fusion protein. CCF4/AM-loaded HeLa cells were infected with WA-314 and WA-314 YopD-ALFA, both carrying pMK-bla encoding for YopE-β-lactamase. Both strains translocated similar amounts of a YopE β-lactamase fusion protein into host cells (Fig 1E) [30–32]. Infection of HeLa cells with *Yersinia* leads to rounding of the cells and this phenomenon is referred to as cytotoxicity. Cytotoxicity is mainly mediated by the action of translocated effectors YopE and YopH and is thus an indicator of a functioning T3SS machinery and translocon. Rounding of HeLa cells was induced to a similar extent by infection with WA314 and WA-314 YopD-ALFA but not with WA314ΔYopD, indicating that translocon-dependent effector transloca-tion into host cells is unaffected in WA-314 YopD-ALFA (Fig 1F). To determine whether bind-ing of NbALFA to translocon-associated YopD-ALFA impairs translocon function, Hela cells were infected with WA-314 YopD-ALFA in the presence of NbALFA. NbALFA had no effect on translocation of the effector YopH, as determined by Western blotting of YopH extracted by digi-tonin lysis from HeLa cells, or on cytotoxicity (S1 Fig). We conclude that insertion of the ALFA tag into YopD and binding of NbALFA do not affect translocon function.

## Fluorescence staining of YopD-ALFA in pre-phagosomes, phagosomes and *Yersinia* cells

In previous work we showed that translocon formation by *Y. enterocolitica* occurs in a specific pre-phagosomal host cell compartment, previously referred to as prevacuole [21,33]. The *Yer-sinia*-containing pre-phagosome is derived from the plasma membrane, enriched with PIP2 and characterized by a narrow connection to the extracellular space, which cannot be passed by large extracellular molecules such as antibodies (MW approx. 150 kDa), but by small mole-cules like streptavidin (MW 53 kDa) [21,33]. We therefore assumed that it should be feasible to stain YopD-ALFA in newly formed translocons by adding fluorophore-labeled NbALFA (MW 15 kDa) to fixed but unpermeabilized WA-314 YopD-ALFA infected cells. To test this notion and further investigate the localization of YopD-ALFA in the course of cell infection, we sequentially stained WA-314 YopD-ALFA infected HeLa cells without permeabilization (NbALFA-635), after permeabilization of the HeLa cell membranes with digitonin (NbALFA-580) and after additional permeabilization of the bacterial membranes with 0.1% Triton X-100 (NbALFA-488) (Fig 2A). In unpermeabilized HeLa cells, patchy fluorescence signals associated with bacteria could be detected (Fig 2A, left), confirming that translocon-associated Yop-D-ALFA can be accessed by extracellularly added NbALFA. In digitonin-permeabilized HeLa cells, additional translocon signals could be found that were not seen in unpermeabilized cells, indicating that these translocons resided in closed phagosomes (Fig 2A, middle). After addi-tional permeabilization of the bacterial membranes with Triton X-100, the intrabacterial pool of YopD-ALFA could be visualized in all bacteria, independent of whether they displayed translocons (Fig 2A, right). The diffuse distribution of intrabacterial YopD-ALFA is in clear contrast to the patchy pattern of translocon-associated YopD-ALFA. To better resolve

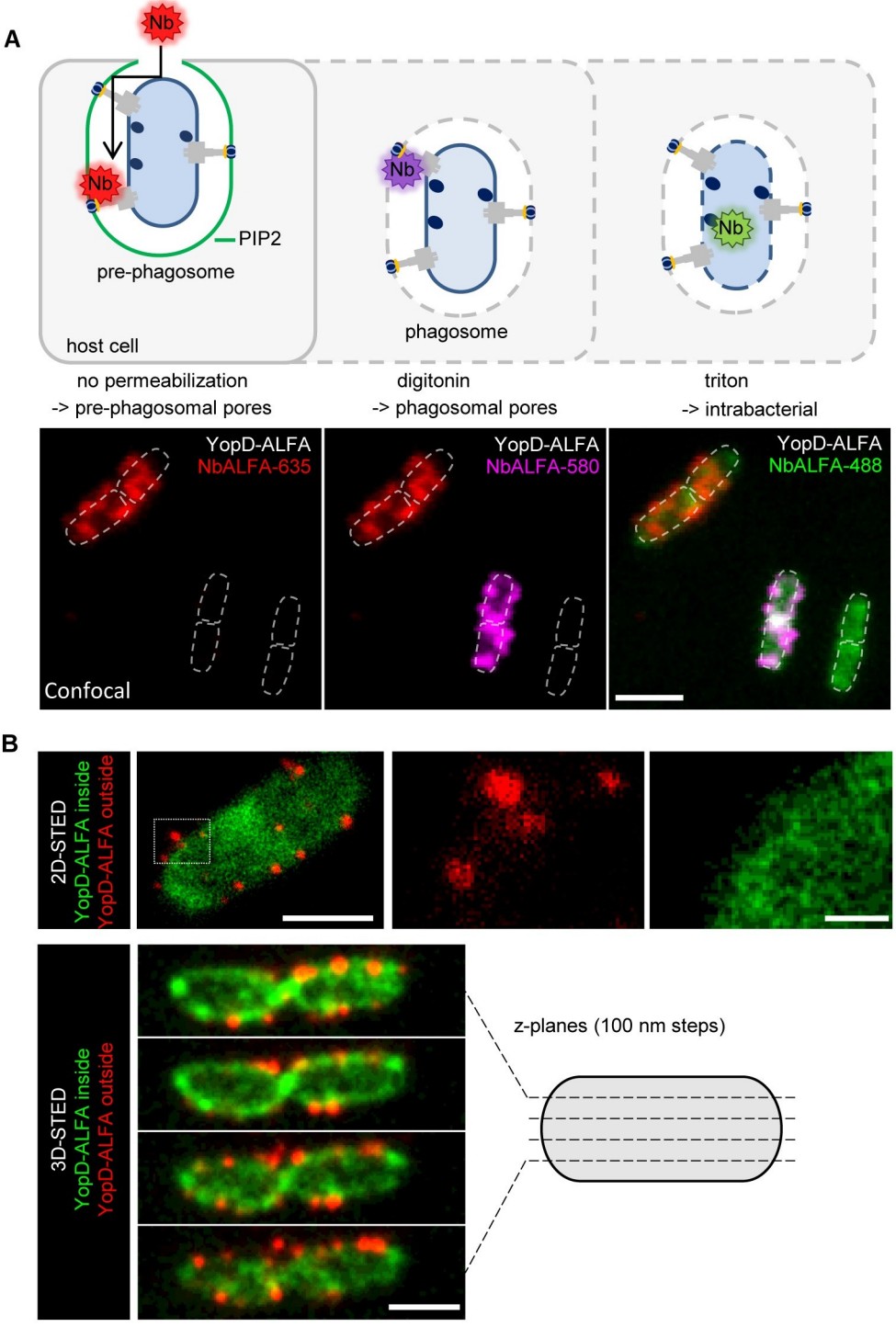

**Fig 2. Differential permeabilization for selective staining of YopD-ALFA in different cellular compartments. (A) Selective nanobody staining of YopD-ALFA in different cellular compartments.** The schematic (top) shows different levels of host- and bacterial cell permeabilization and according accessibility of different pools of YopD-ALFA for NbALFA staining. Rac1Q61L expressing HeLa cells were infected with WA-314 YopD-ALFA at an MOI of 10 for 1 h, fixed and stained with NbALFA-635 without prior permeabilization to specifically target translocon associated YopD-ALFA in the pre-phagosomal compartment (left, shown in red). Host cell membranes were permeabilized with digitonin and translocons located in closed phagosomes were stained with NbALFA-580 (middle, shown in magenta). Note that pre-phagosomal YopD-ALFA was already saturated with NbALFA-635 (red) during the first staining step. Finally, also the bacterial membranes were permeabilized with triton and the intrabacterial pool of YopD-ALFA was

stained with NbALFA-488 (right, shown in green). Scale bar: 5 μm. **(B) 2D and 3D-STED imaging of intrabacterial and translocon-associated YopD-ALFA.** Rac1Q61L expressing HeLa cells were infected with WA-314 YopD-ALFA at an MOI of 10 for 1 h, fixed and stained with NbALFA-635 (shown in red) with prior permeabilization of host cell membranes using digitonin to target translocon associated YopD-ALFA. Bacterial membranes were permeabilized with triton and the intrabacterial pool of YopD-ALFA was stained with NbALFA-580 (shown in green). The images were acquired using super resolution STED microscopy. The boxed region in the left of the image is depicted as enlargements in separate channels at the side. Scale bars: 1 μm (2D-STED overview and 3D STED) and 200 nm (2D-STED enlargements).

translocon associated from intrabacterial YopD-ALFA, we employed super resolution 2D and 3D STED microscopy (Fig 2B). While the intrabacterial distribution of YopD-ALFA remained diffuse also at this level of resolution, the extrabacterial YopD-ALFA produced distinct signals with a lateral extent of about 40 nm, which previously were identified as single translocons [21]. Overall, differential YopD-ALFA staining allows to visualize *Yersinia* translocons located in pre-phagosomes and phagosomes, as well as the intrabacterial YopD pool.

## Live imaging of *Yersinia* translocon formation during cell infection

Given the ability to stain translocons in fixed and unpermeabilized cells by external addition of fluorophore-labeled NbALFA, we hypothesized that this may also enable the recording of translocon formation in living cells. To test this possibility, live HeLa cells expressing GFP-LifeAct and myc-Rac1Q61L were infected with WA-314 YopD-ALFA in the presence of NbALFA-580 and imaged using spinning disc microscopy with one acquisition per minute. myc-Rac1Q61L was overexpressed in all Hela cell live imaging experiments because it strongly increases the percentage of bacteria forming translocons (S2 Fig) by stimulating uptake of the bacteria into the pre-phagosome [21]. GFP-LifeAct was expressed to visualize host cells and to enable the localization of the cell adhering bacteria. With this approach, appearance and disappearance of fluorescence signals corresponding to translocon-associated YopD-ALFA could be recorded over time (representative event in Fig 3A and S1 Movie). From their first visible appearance (at 5 min in Fig 3A), the number and intensity of YopD-ALFA fluorescence signals peaked after about 20 min and decreased thereafter. The mean overall lifespan of YopD-ALFA fluorescence signals, defined as their first visible appearance until their complete vanishing, was determined to be 26.6 +/- 13 min (mean +/- S.D., Fig 3B). The disappearance of the fluorescence signals was most certainly not due to photo bleaching because no decay of fluorescence was observed in recordings with considerably higher imaging frequency (e.g. acquisition rate: 3 per min in Fig 3C vs. 1 per min in Fig 3A). To further evaluate potential photobleaching, HeLa cells were infected with WA-314 YopD-ALFA and fixed with paraformaldehyde. Fixed YopD-ALFA was stained with NbALFA-580 and subjected to live imaging employing the same conditions as for NbALFA-580-stained native Yop-D-ALFA. The fixed YopD-ALFA fluorescence signal decreased by 18% within 105 min, as compared to a 74% decrease of the signal in a representative live imaging experiment during 40 min (S3 Fig). This indicates that only minor photobleaching occurs during the live imaging time period und that a decrease of the fluorescence signal most likely is caused by degradation or dissolution of YopD-ALFA. Taken together, for the first time we filmed assembly and disassembly or degradation of T3SS translocons in living host cells, thus providing new insights into the spatiotemporal dynamics of this central T3SS activity.

## Spatiotemporal dynamics and sequence of uptake of *Yersinia* into the pre-phagosome followed by translocon formation

The ability to film translocons allowed us to study their spatiotemporal correlation with uptake of the bacteria into the pre-phagosome [21,33]. For this, we infected HeLa cells expressing the

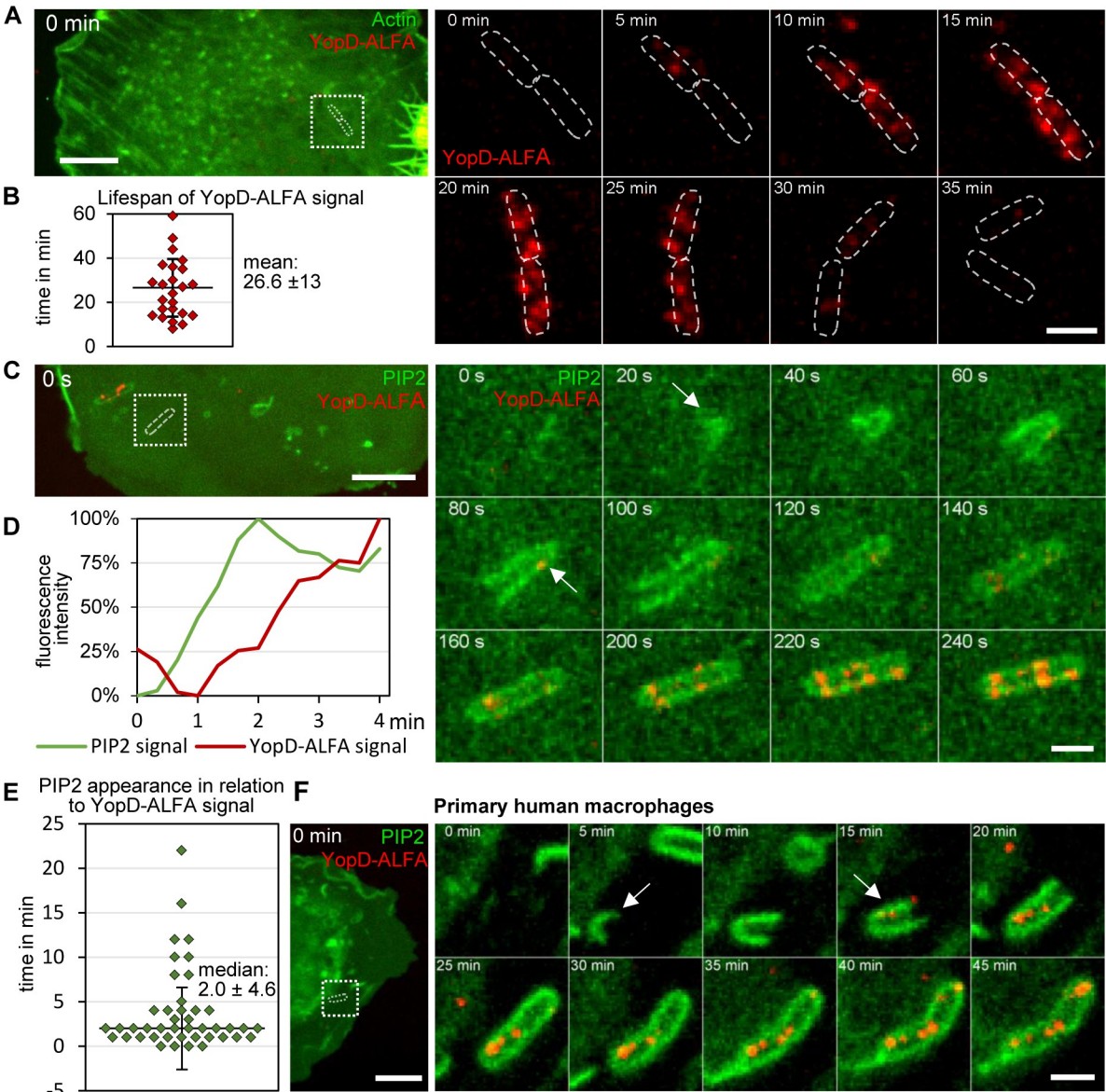

**Fig 3. Nanobody-based live imaging of translocons: Formation and lifespan of the translocon during cell infection. (A) Live imaging of translocons during HeLa cell infection.** HeLa cells expressing myc-Rac1Q61L and GFP-LifeAct were infected with WA-314 YopD-ALFA at an MOI of 20 and incubated with NbALFA-580 diluted in the cell culture medium. Cells were imaged with a spinning disk microscope recording z-stacks every minute. Stacks for each time point were combined to one image using maximum intensity projection and one image every 5 min is shown. The left panel shows the overview image at 0 min. The boxed region in the overview image shows the area of the video depicted in still frames to the right. Dashed white lines indicate the outline of the bacteria. Scale bars: 10 μm (overview) and 2 μm (still frames). **(B) Lifespan of the translocon.** The lifespan of the translocons was determined using movies that recorded the YopD-signal of individual bacteria from their formation to disappearance. Experimental conditions are as in (A). n = 25 bacteria (7 independent experiments, 13 host cells) **(C) Engulfment in PIP2-positive membranes precedes translocon formation in HeLa cells.** HeLa cells expressing myc-Rac1Q61L and PLCδ1-PH-GFP were infected with WA-314 YopD-ALFA at an MOI of 20 and incubated with NbALFA-580 diluted in cell culture medium. Cells were imaged with a spinning disk microscope recording z-stacks every 20 s. Stacks for each time point were combined to one image using maximum intensity projection and one image every 20 s is shown. The left panel shows the overview image at 0 min. The boxed region in the overview image shows the area of the video depicted in still frames to the right. White arrows indicate the appearance of PLCδ1-PH-GFP and the first translocon signal. Scale bar: 10 μm (overview) and 2 μm (still frames). **(D) Fluorescence intensities of PIP2 marker PLCδ1-PH-GFP and YopD-ALFA signals.** The relative fluorescence intensities of PLCδ1-PH-GFP and NbALFA-580 signals at the bacteria in (C) were plotted to illustrate the temporal relationship of signal appearances. **(E) Temporal relationship of engulfment in PIP2-positive membranes and appearance of YopD-ALFA signal.** The time intervals between first occurrence of the PLCδ1-PH-GFP and first YopD-ALFA signals were measured based on live imaging experiments performed as in (C). Each dot represents one measurement. n = 43 bacteria (3 independent experiments, 13 movies). **(F) Engulfment in PIP2-positive membranes precedes translocon formation in primary human macrophages.** Primary human macrophages expressing PLCδ1-PH-GFP

were infected with WA-314 YopD-ALFA at an MOI of 20 and incubated with NbALFA-580 diluted in cell culture medium. Cells were imaged with a spinning disk microscope recording z-stacks every minute. Stacks for each time point were combined to one image using maximum intensity projection and one image every 5 min is shown. The left panel shows the overview image at 0 min. The boxed region in the overview image shows the area of the video depicted in still frames to the right. White arrows indicate the appearance of PLCδ1-PH-GFP and the first translocon signal. Scale bars: 10 μm (overview) and 2 μm (still frames).

PIP2 sensor PLCδ1-PH-GFP with WA-314 YopD-ALFA in the presence of NbALFA-580 and performed live cell imaging (one z-stack every 20 s). Still frames of a representative event are depicted in Fig 3C, showing a bacterium being completely enclosed with PIP2 positive host membranes. Recruitment of PLCδ1-PH-GFP started at one pole of the bacterial cell und then continued until the whole cell was encompassed (Fig 3C). In the representative example, the time to complete encompassment of the bacterial cell was around 100 s (start and completion of recruitment at 20 s and 120 s, respectively; Fig 3C). In this example the first YopD-ALFA signal (at 80 s; Fig 3C) was observed about 60 s after the first PLCδ1-PH-GFP signal and occurred at the pole of the bacterium that was engulfed first by PLCδ1-PH-GFP (Fig 3C and 3D). Thereafter, the number and intensity of the YopD-ALFA translocon signals further increased until the 240 s time point (Fig 3C and 3D; S2 Movie). The median time lag between the first visible PLCδ1-PH-GFP and YopD-ALFA signals was determined to be 2.0 +/- 4.6 min (median +/- S.D., Fig 3E). Importantly, in all cases examined, the YopD-ALFA translocon signals occurred after or at the earliest simultaneously with the accumulation of PLCδ1-PH-GFP around the bacteria (Fig 3E). To test the spatiotemporal coordination of PIP2 accumulation and translocon formation in physiological target cells of pathogenic yersiniae, we employed primary human macrophages [34]. Live macrophages expressing PLCδ1-PH-GFP were infected with WA-314 YopD-ALFA in the presence of NbALFA-580 and investigated with live cell imaging. A representative movie shows that also in the macrophages the bacteria were enclosed with PLCδ1-PH-GFP positive membrane (at 5 min in Fig 3F; S3 Movie) before the YopD-ALFA translocon signals appeared (at 15 min in Fig 3F; S3 Movie). In summary, these data demonstrate a close spatiotemporal sequence of bacterial uptake into the pre-phagosome followed by T3SS translocon formation. This infers, but does not prove, that pre-phagosomal membrane composition or another host factor in this compartment triggers translocon formation.

## Spatiotemporal dynamics of galectin-3 and GBP-1 recruitment to *Yersinia* containing phagosomes harboring translocons

It has previously been shown that the T3SS of *Yersinia* can damage the phagosome membranes surrounding these bacteria in cells [35,36]. Yet, the dynamics and spatiotemporal relationship between membrane damage and formation of translocons/pores, that are thought to induce the membrane disruption, have not been elucidated. GFP-galectin-3 has been used as a sensor for membrane damage because of its ability to attach to β-galactose-containing glycoconjugates present in the luminal leaflet of phagosomal membranes (Fig 4A) [37]. When expressed in the cytosol of host cells, GFP-galectin-3 accumulates at phagosomal membranes when these have been ruptured e.g., by T3SS translocons (Fig 4A). To first confirm that membranes are disrupted by the *Yersinia* T3SS, we infected HeLa cells expressing GFP-galectin-3 and myc-Rac1Q61L with different *Yersinia* strains (Table 1). We observed recruitment of GFP-galectin-3 by approximately 13% of both, cell-associated wild type WA-314 YopD-ALFA and effector-deficient WA-C pTTSS, but not by T3SS-deficient WA-C bacteria (Fig 4B). This confirms that phagosome disruption is caused by the *Yersinia* T3SS without the involvement of effectors.

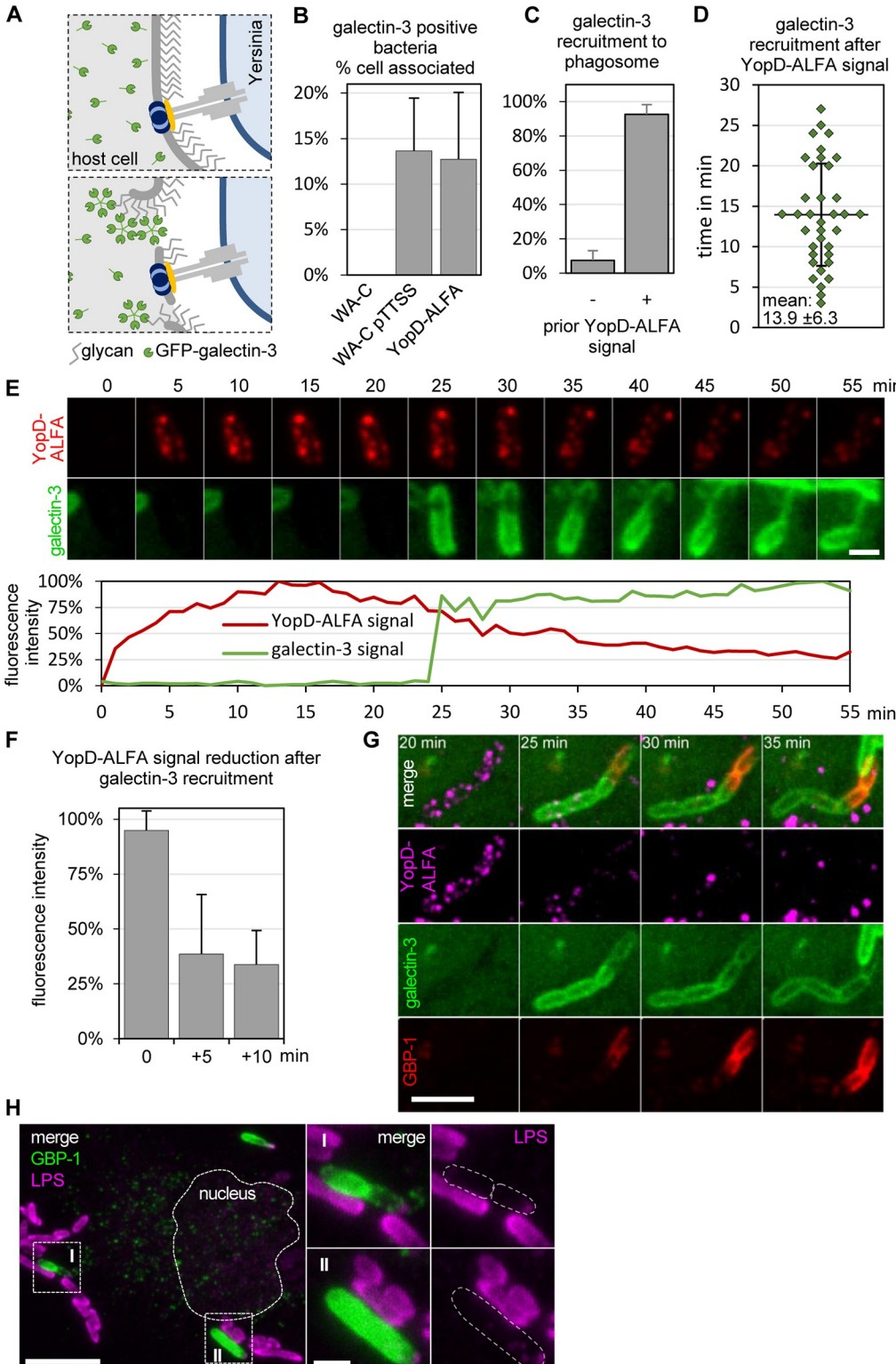

**Fig 4. Nanobody-based live imaging of translocons: Galectin-3 and GBP-1 recruitment upon translocon induced membrane damage. (A)** Schematic representation of galectin-3 recruitment following membrane damage during infection. Galectin-3 (shown in green) is found in the cytosol of the host cell. Translocon formation appears to induce membrane damage allowing access of galectin-3 to glycans in the lumen of vacuoles. **(B) Vacuolar membrane damage by the T3SS.** HeLa cells expressing myc-Rac1Q61L and GFP-galectin-3 were infected with WA-C, WA-C pTTSS and

WA-314 YopD-ALFA at an MOI of 100 for 1 h, fixed and permeabilized using digitonin. Cells were stained with anti-YopD antibody, Alexa633 phalloidin and DAPI. The percentage of galectin-3 positive bacteria per cell was quantified for WA-C, WA-C pTTSS and WA-314 YopD-ALFA (n = 1680, 9 host cells; n = 508 bacteria, 7 host cells; n = 1065 bacteria, 10 host cells). Only cells harboring translocon forming bacteria were analyzed for WA-C pTTSS and WA-314 YopD-ALFA infections. **(C) Fraction of galectin recruitments without and with prior YopD-ALFA signal.** HeLa cells expressing myc-Rac1Q61L and GFP-galectin-3 were infected with WA-314 YopD-ALFA at an MOI of 20 and incubated with NbALFA-580 diluted in cell culture medium. Cells were imaged with a spinning disk microscope recording z-stacks every minute. Galectin-3 recruitment events were quantified with respect to whether YopD-ALFA signal is present before recruitment. n = 330 uptake events (6 independent experiments, 38 movies). **(D) Temporal relationship of YopD-ALFA signal appearance and galectin-3 recruitment.** HeLa cells expressing myc-Rac1Q61L and GFP-galectin-3 were infected with WA-314 YopD-ALFA at an MOI of 20 and incubated with NbALFA-580 diluted in cell culture medium. Cells were imaged with a spinning disk microscope recording z-stacks every minute. The time intervals between first occurrence of the translocon signal and first GFP-galectin-3 signals were measured. Each dot represents one measurement. n = 36 bacteria (4 independent experiments; 9 movies). **(E) Galectin-3 recruitment to phagosomes containing translocon forming bacteria.** Live imaging experiments were performed as in (C). The z-stacks for each time point were combined to one image using maximum intensity projection and one image every 5 min is representatively shown. Scale bar: 2 μm. The relative fluorescence intensities at the bacteria were plotted to illustrate the temporal relationship of NbALFA-580 and GFP-galectin-3 signals. **(F) Loss of YopD-ALFA signal after GFP-galectin-3 recruitment.** Live imaging experiments were performed as in (C). The relative fluorescence intensities of the translocon signal were measured in the last frame before and in the frames 5 min and 10 min after recruitment of GFP-galectin-3. n = 4 measurements (1 experiment, 3 host cells). **(G) GBP-1 recruitment to galectin-3 positive bacteria.** HeLa cells expressing myc-Rac1Q61L, mScarlet-galectin-3 (shown in green) and GFP-GBP-1 (shown in red) were infected with WA-314 YopD-ALFA at an MOI of 20 and incubated with NbALFA-580 (shown in magenta) diluted in cell culture medium. Cells were imaged with a spinning disk microscope recording z-stacks every 5 minutes. The z-stacks for each time point were combined to one image using maximum intensity projection and images are shown starting 20 min after uptake of the bacteria. Scale bar: 5 μm. **(H) GBP-1 positive bacteria lack LPS antibody staining.** HeLa cells expressing myc-Rac1Q61L and GFP-GBP-1 (shown in green) were infected with WA-314 YopD-ALFA at an MOI of 30, fixed and permeabilized using digitonin. Cells were stained with anti-LPS antibody (shown in magenta). The boxed regions (I, II) in the overview image are depicted as enlargements in separate channels to the right. Dashed white lines indicate the outline of the nucleus in the overview image and the bacteria in the enlargements. Scale bar: 10 μm (overview) and 2 μm (enlargement).

To test the spatiotemporal relation of translocon formation and membrane damage, we infected HeLa cells expressing GFP-galectin-3 with WA-314 YopD-ALFA in the presence of NbALFA-580 and performed live cell imaging. This revealed that GFP-galectin-3 recruitment was preceded by detectable YopD-ALFA signals in 92.6% +/- 5.6% (mean +/- S.D.) of all GFP-galectin-3 recruitment events (Fig 4C). The mean time interval between translocon formation and GFP-galectin-3 recruitment was determined to be 13.9 +/-6.3 min (mean +/- S.D., Fig 4D). We also noticed that GFP-galectin-3 was recruited relatively abruptly to the entire phagosomal membrane and that its recruitment was accompanied by a decrease in the YopD-ALFA translocon signal (n = 4; Fig 4E and 4F and S4 Movie). After we excluded that photobleaching

**Table 1. *Yersinia enterocolitica* strains.**

| Strain | Relevant characteristic | Source/ References |
|---|---|---|
| WA-314 | wild type strain carrying virulence plasmid pYV; serogroup O8; kanamycin resistance cassette in non-coding region of pYV-O8 | [51,53] |
| WA-C | pYV-cured derivative of WA-314 | [53] |
| WA-C pTTSS | WA-C harboring pTTSS encoding the TTSS secretion/translocation apparatus of WA-314 but no Yop effector genes; Spt$^R$ | [54] |
| WA-314ΔYopD | WA-C harboring pYVΔyopD; Kan$^R$ | [21] |
| WA-314 YopD-ALFA | WA-314 with ALFA-tag inserted in YopD; Kan$^R$ | this study |
| WA-314 pMK-bla | WA-314 harboring pMK-bla containing YopE53-β-lactamase fusion; Kan$^R$, CM$^R$ | [31] |
| WA-314 YopD-ALFA pMK-bla | WA-314 YopD-ALFA harboring pYopE-bla; Kan$^R$, CM$^R$ | this study |

causes relevant reduction in YopD-ALFA/NbALFA fluorescence signals in our experiments (S3 Fig), we wanted to obtain alternative evidence for YopD degradation after GFP-galectin-3 recruitment. For this, we compared and quantified NbALFA- and antibody-mediated fluorescence stainings of YopD-ALFA and YopD, respectively, in fixed cells. The results complement the live imaging data and very clearly show that both, NbALFA- and antibody-mediated fluorescence signals of translocons diminish or disappear at bacteria surrounded by GFP-galectin-3 (S4 Fig). These results suggest that the translocon protein YopD is degraded or disassembled in association with galectin-3 recruitment but do not formally prove that galectin-3 is functionally involved in YopD disappearance.

Recently it was shown that galectin-3 promotes recruitment of the guanylate-binding proteins (GBPs) GBP-1 and GBP-2 to *Yersinia*-containing compartments [35]. GBPs belong to the family of interferon-inducible GTPases and facilitate cell-intrinsic immunity by targeting host defense proteins to pathogen containing compartments [38]. Further, GBP-1 was recently shown to directly bind to LPS of Gram-negative bacteria and function as an LPS-clustering surfactant that disrupts the physicochemical properties of the LPS layer [39–41]. To investigate the spatiotemporal coordination of galectin-3 and GBP-1 recruitment to bacteria that formed translocons, HeLa cells co-expressing galectin-3-mScarlet and GFP-GBP-1 were infected with WA-314 YopD-ALFA in the presence of NbALFA-580 and subjected to live cell imaging (Fig 4G; S5 Movie). GFP-GBP-1 was recruited specifically to bacteria that previously had formed translocons and recruited GFP-galectin-3 (100% of observed GFP-GBP-1 recruitment events, n = 45). GFP-GBP-1 recruitment (to bacteria that had formed translocons) was regularly observed shortly after or concomitant with GFP-galectin-3 recruitment (Fig 4G; S5 Movie). Staining with an anti-O8 LPS antibody failed to stain GBP-1 coated bacteria suggesting that GBP-1 interacts with the *Yersinia* LPS and hinders detection by the anti-LPS antibody (Fig 4H). These data suggest that *Yersinia* translocons cause disruption of phagosomal membranes which leads to sequential galectin-3 and GBP-1 recruitment and, likely through the recruitment of additional factors, to translocon degradation or disassembly.

## Discussion

Here we used live cell imaging to characterize the spatiotemporal sequence of molecular events associated with *Yersinia* translocon assembly and disassembly in host cells. To this end, we constructed a *Yersinia* strain expressing an ALFA-tag labeled YopD that retained its functionality and could be visualized in fixed and living cells by binding to a fluorescently labeled nanobody (NbALFA). Live cell imaging was also facilitated by the fact that *Yersinia* translocon formation occurs in a pre-phagosomal host cell compartment where YopD-ALFA is accessible by externally added NbALFA. Thus the incorporation of YopD-ALFA into translocons, as measure for translocon formation, could be recorded and temporally and spatially correlated with the accumulation of the following biosensors i) PLCδ1-PH-GFP, which senses PIP2 at the pre-phagosome; ii) GFP-galectin-3, which senses phagosomal membrane disruption; and iii) GFP-GBP1, which senses activation of the cell autonomous immune system at the phagosome.

Translocon formation was always initiated seconds to a few minutes after engulfment of the bacteria in PIP2 positive membranes of the pre-phagosome, suggesting that a specific host cell factor in the pre-phagosomes may trigger the secretion of translocon proteins by the T3SS and/or is required for their membrane integration. PIP2 rich pre-phagosomes may also recruit host cell receptors like FPR1 and CCR5, which have been reported to promote *Y. pestis* and *Y. pseudotuberculosis* translocon formation [15,42].

Disruption of the phagosomal membrane by *Yersinia* required a functional T3SS, was associated with only a small fraction of cell-associated bacteria on which we previously detected

translocons in >90%, and occurred with a delay of approximately 14 min after translocon formation. This indicates that membrane integration of the translocons per se is not sufficient for membrane disruption and subsequent entry of galectin-3. It remains to be elucidated how translocons compromise the phagosome membrane, e.g., whether they act in terms of unregulated translocon activity if they are separated from the T3SS during phagosome maturation or whether membrane disrupting host immune factors that recognize the translocon might be involved.

In a recent study galectin-3 was found to recruit the guanylate binding proteins (GBPs) GBP-1 and GBP-2 to *Yersinia* containing vacuoles dependent on a functional T3SS [35]. GBPs belong to the large group of interferon induced antimicrobial host cell factors known to be recruited to pathogen-containing vacuoles (PVs). They escort antimicrobial factors to the PVs and thereby contribute to the cell-autonomous immunity [38,43]. Of note, the galectin-3 signal in our study usually covered the whole circumference of the bacteria indicating that the membrane is not substantially ruptured or detached from the bacteria as described for *Shigella* [37,44]. GFP-galectin-3 and GBP1-GFP were sequentially recruited to bacteria-containing phagosomes that previously had formed translocons and were associated with a reduction of the YopD-ALFA and LPS signals. The exact mechanisms responsible for the obvious dissolution of the translocon and the bacterial cell membranes are not known. Future experiments involving galectin-3 or GBP1 knockout cell lines are needed to investigate whether these factors are involved in pathways leading to translocon degradation.

In summary, we here describe a new method for visualizing and filming the assembly and disassembly of *Yersinia* translocons by fluorescence microscopy in living cells. In this way, key aspects of the dynamics of translocon formation, its effects on membrane integrity, and recognition by host cell defense mechanisms could be recorded with high spatiotemporal resolution. The described approach might also be valuable for imaging of translocator proteins in other bacterial species. For this it will be critical to insert the tag into positions in the different translocators so that no interference with translocon function occurs.

More highly resolved molecular details of T3SS translocon formation, the effects of translocon pores on host membranes and their recognition by host immune factors may become available through the development and application of super resolution live imaging technologies like live cell STED and MINFLUX microscopy [45].

## Materials and methods

### Ethics statement

Approval for the analysis of anonymized blood donations (WF-015/12) was obtained by the Ethical Committee of the Ärztekammer Hamburg (Germany).

### Materials

All standard laboratory chemicals and supplies were purchased from Roth (Karlsruhe, Germany), Sigma-Aldrich (Steinheim, Germany) or Merck (Hohenbrunn, Germany) unless indicated otherwise.

### Plasmids

The following plasmids were described previously: PLCδ1-PH-GFP [46] was provided by Tamás Balla (National Institutes of Health, Bethesda, MD). The myc-Rac1Q61L plasmid [47] was kindly provided by Dr. Pontus Aspenström (Uppsala University, Uppsala, Sweden) and the GFP-GBP-1 plasmid [41] by P. Broz (University of Lausanne, Epalinges, Switzerland).

pEGFP-galectin-3 [48] was a gift from Tamotsu Yoshimori (Addgene plasmid # 73080) and the mScarlet-galectin 3 was generated by using the pEGFP-galectin-3 plasmid and replacing eGFP by mScarlet at the NheI/BglII sites. pCMV-NbALFA-mScarlet-I (NanoTag Biotechnologies, Germany) was used as PCR amplification template (mScarlet fwd NheI: AGATCCGC-TAGCGATGGTGAGCAAGGGCGAG; mScarlet rev BglII: TGCCATAGATCTCTTGTACAGCTCGTCCAT). The plasmid pMK-bla [49] was kindly provided by Erwin Bohn (Institute of Medical Microbiology and Hygiene, University of Tuebingen, Tuebingen, Germany) and the Lifeact-eGFP construct [50] was a kind gift of Michael Sixt (Max Planck Institute for Biochemistry, Munich, Germany).

## Antibodies and nanobodies

Polyclonal rabbit anti-YopB (aa 1–168) and anti-YopD (aa 150–287) as well as rat anti-YopB (aa 1–168) antibodies were produced as described previously [21]. Rabbit polyclonal anti-*Y. enterocolitica* O:8 was purchased from Sifin (Berlin, Germany). Secondary anti-IgG antibodies and their sources were: Alexa488 chicken goat anti-rat, Alexa568 goat anti-rabbit, Alexa647 goat anti-rabbit, (Molecular Probes, Karlsruhe, Germany), horseradish peroxidase linked donkey anti-rabbit (GE Healthcare, Chicago, USA). Fluorescently labeled primary camelid anti-ALFA nanobodies (NbALFA) and their source were: Alexa Fluor 488 FluoTag-X2 (NbALFA-488), Alexa Fluor 580 FluoTag-X2 (NbALFA-580), AbberiorStar635P FluoTag-X2 (NbALFA-635) (NanoTag Biotechnologies, Göttingen, Germany).

## Source and generation of *Yersinia* mutants

The *Yersinia* strains used here are listed in Table 1. *Y. enterocolitica* wild type strain WA-314 was a gift of Jürgen Heesemann (Max von Pettenkofer Institute, Munich, Germany) and described previously [51]. WA-314 YopD-ALFA was generated using a CRISPR-Cas12a-assisted recombineering approach [52]. In brief, a double stranded Homology Directed Repair (HDR) fragment containing the ALFA-tag and linker sequence was generated via overlap extension PCR. For this a 500 bp homology arm (HomA) was amplified from the *Y. enterocolitica* pYV virulence plasmid with the reverse primer YopD-ALFA HomA rev including part of the ALFA-tag insert and linker and the corresponding forward primer YopD-ALFA HomA fwd. The other homology arm (HomB) was amplified using the forward primer YopD-ALFA HomB fwd including the remaining part of the ALFA-tag insert and linker and the corresponding reverse primer YopD-ALFA HomB rev. Both homology arms were used as templates in an overlap extension PCR using the outer primers (YopD-ALFA HomA fwd and YopD-ALFA HomB rev) to generate the final HDR fragment.

The crRNAs required for targeting Cas12a to the defined insertion site were designed based on the 20 bp protospacer following the 3'-end of a PAM (5'-TTN-3'). The respective oligonucleotides were designed with Eco31L overhangs at the 5'- and 3'-ends (YopD-crRNA fwd and YopD-crRNA rev), annealed and ligated into the Eco31L digested pAC-crRNA vector harboring also a sacB sucrose sensitivity gene. 700 ng of the HDR fragment and 350 ng of the pAC-crRNA were electroporated into an electrocompetent WA-314 strain carrying pKD46-Cas12a, which harbors the lambda Red recombinase under control of an arabinose inducible promotor, Cas12a (Cas12a/Cpf1 from *Francisella novicida*) and a temperature-sensitive replicon. After successful editing of the virulence plasmid, the pAC-crRNA and pKD46-Cas12a plasmids were cured from the bacteria. Correct insertion of the ALFA-tag was confirmed by PCR and sequencing. The editing resulted in the expression of a modified YopD carrying the ALFA-tag between amino acids 194 and 195 (GGSGGSGGP**SRLEEELRRRLTE**PGGGGS; linker, **ALFA-tag**, linker).

**Table 2. Oligonucleotides and sequences.**

| | |
|---|---|
| YopD-ALFA HomA fwd | TATTATCCTAACTTATTATTTTTAATTTAATAATAAAAGCCCTGGATTACCATTAGTTAA |
| YopD-ALFA HomA rev | TTGGAAGAGGAACTGAGACGCCGCTTAACTGAACCAGGCGGAGGTGGATCTATCGGGAGAATATGGAAACCAGA |
| YopD-ALFA HomB fwd | GCGGCGTCTCAGTTCCTCTTCCAAACGGCTCGGGCCACCAGACCCGCCCGAACCACCATCCTCTCTGCTTACCGCTTTAT |
| YopD-ALFA HomB rev | AAAGCGGTGAGGTTAAAAAAA |
| YopD-crRNA fwd | TAGATCATATTCTCCCGATATCCTC |
| YopD-crRNA rev | AGACGAGGATATCGGGAGAATATGA |
| final insert sequence (<u>linker</u>, **ALFA-tag**, <u>linker</u>) | <u>GGTGGTTCGGGCGGGTCTGGTGGCCCG</u>**AGCCGTTTGGAAGAGGAACTGAGACGCCGCTTAACTGAA**<u>CCAGGCGGAGGTGGATCT</u> |

## Oligonucleotides and sequences

The *oligonucleotides* and sequences used are listed in Table 2.

## Cell culture and transfection

HeLa cells (ACC#57, DSMZ-German Collection of Microorganisms and Cell Cultures) were cultured at 37˚C and 5% CO2 in DMEM (Invitrogen, GIBCO, Darmstadt, Germany) supplemented with 10% FCS (v/v). For infection with bacteria, HeLa cells were seeded in 6 well plates ($3x10^5$ cells per well) or on glass coverslips (6x104 cells per well; confocal: Precision coverslips, round, 12 mm diameter, No 1.5, with precision thickness, Hartenstein, Würzburg, Germany; STED: 12mm, No. 1.5H for high resolution, Marienfeld GmbH, Lauda-Königshafen, Germany). For live imaging $2.5 \times 10^4$ HeLa cells were seeded in ibidi μ-slide 8 wells (ibidi, Martinsried, Germany). HeLa cells were transfected with 0.25 μg plasmid for coverslips or 0.125 μg plasmid for 8 well slides using Turbofect (Thermo Fisher Scientific, Waltham, Massachusetts, USA) for 16 h according to the manufacturer's protocol.

Human peripheral blood monocytes were isolated from heparinized blood as described previously [55]. Monocytes/Macrophages were cultured in RPMI1640 (Invitrogen) containing 20% heterologous human serum (v/v) for 7 days with medium changes every three days. Macrophages were transfected with the Neon Transfection System (Invitrogen) with 5 μg DNA per $10^6$ cells (1000 V, 40 ms, 2 pulses) and infected 4 h after transfection.

## Preparation of bacteria

*Yersinia* were grown in lysogeny broth (LB) supplemented with nalidixic acid, kanamycin, spectinomycin or chloramphenicol as required at 27˚C overnight and then diluted 1:20 in fresh LB broth, followed by cultivation at 37˚C for 1.5 h to induce expression of the T3SS. For cell infection, bacteria were centrifuged, resuspended in ice-cold PBS and added to target cells at a defined multiplicity of infection (MOI), as specified in the figure captions. Bacteria were then centrifuged at 200 x *g* for 1 min onto the target cells to synchronize the bacterial attachment. For in-vitro Yop secretion, EGTA (5 mM), MgCl2 (15 mM) and glucose (0.2%, w/v) was added to the growth medium for Ca2+ chelation after 1.5 h at 37˚C, followed by another 3 h of incubation at 37˚C, as described before [54]. The resulting samples were analyzed by SDS-PAGE, followed by either Coomassie staining or transfer to a PVDF membrane (Immobilon-P, Millipore), and analysis by Western blot using antisera against YopB and YopD.

## Fluorescence labeling

Infected cells were washed twice with PBS and fixed with 4% PFA (v/v; Electron Microscopy Science, Hatfield, USA) in PBS for 5 min. Samples were treated with digitonin solution (90 µg/mL in PBS) to permeabilize cellular membranes and allow access of the nanobody to translocon-associated YopD-ALFA. For antibody stainings or staining of intrabacterial YopD-ALFA using the nanobody, samples were permeabilized with 0.1% Triton X-100 (v/v) in PBS for 15 min. After fixation and permeabilization coverslips were washed twice with PBS. Unspecific binding sites were blocked with 3% bovine serum albumin (BSA, w/v) in PBS for at least 30 min. Samples were then incubated with the indicated primary antibody (1:50) or fluorescently labeled FluoTag-X2 anti-ALFA nanobody (1:200) for 1 h (16 h for STED samples using the nanobody) and incubated with a 1:200 dilution of the suitable fluorophore-coupled secondary antibody or fluorophore-coupled phalloidin (1:200, Invitrogen) and 4',6-diamidino-2-phenylindole (DAPI; 300 nM, Invitrogen) as indicated for 45 min. Nanobodies as well as primary and secondary antibodies were applied in PBS supplemented with 3% BSA. After each staining coverslips were washed three times with PBS. Coverslips for confocal microscopy were mounted in ProLong Diamond (Thermo Fisher Scientific) while STED samples were mounted in ProLong Gold (Thermo Fisher Scientific, Waltham, USA).

## Confocal microscopy

Fixed samples were analyzed with a confocal laser scanning microscope (Leica TCS SP8) equipped with a 63x oil immersion objective (NA 1.4) and Leica LAS X SP8 software (Leica Microsystems, Wetzlar, Germany) was used for acquisition.

## Live cell imaging

For live imaging the cells were seeded in 8 well slides and were placed in the prewarmed chamber supplied with 5% CO2 of the spinning disc microscope Visitron SD-TRIF (Nikon Eclipse TiE, Nikon, Japan) with a 63x oil immersion objective (NA 1.40) and the VisiView software (Visitron Systems, Germany). The nanobody was diluted 1:200 in 200 µl DMEM with 10% FCS and mixed with the WA-314 YopD-ALFA. The number of bacteria was chosen according to the intended MOI. The medium was removed from the cells and the 200 µl medium containing nanobody and bacteria was added. The imaging process was started immediately.

## Super resolution imaging

STED nanoscopy and corresponding confocal microscopy were carried out in line sequential mode using an Abberior Instruments Expert Line STED microscope based on a Nikon Ti-E microscopy body and employed for excitation and detection of the fluorescence signal a 60x Plan APO 1.4 oil immersion objective. A pulsed 640 nm laser was used for excitation and a pulsed near-infrared laser (775 nm) was used for STED. The detected fluorescence signal was directed through a variable sized pinhole (1 Airy unit at 640 nm) and detected by avalanche photo diodes (APDs) with appropriate filter settings for Cy5 (615–755 nm). Images were recorded with a dwell time of 0.5 µs and the pixel size was set to be 10 nm for 2D-STED or the voxel size was set to 40x40x50 nm for 3D-STED. The acquisitions were carried out in time gating mode i.e. with a time gating delay of 750 ps and a width of 8 ns. 3D-STED images were acquired with 80% 3D-STED donut.

## Image analysis

The z-stacks of images acquired of both fixed and live samples were combined to one image using maximum intensity projection. These images were used to determine the lifespan of a detectable translocon signal, the fluorescence intensity of PLCδ1-PH and YopD-ALFA signal in a region of interest around the bacterium for each time point as well as the time PLCδ1-PH is present before translocon formation. In addition, the percentage of galectin-3 positive bacteria in cells harboring translocon forming bacteria was quantified in fixed samples. Live imaging was used to determine the number of galectin-3 positive bacteria with and without prior YopD-ALFA signal as well as the time interval between translocon formation and galectin-3 recruitment. The fluorescence intensity of YopD-ALFA and GFP-galectin-3 signal was measured in a region of interest around the bacterium for each time point of a representative movie. In general, the fluorescence intensity measurements were normalized to the lowest intensity measured and the highest intensity value was set to 100%. Live imaging data were additionally analyzed for the presence of galectin-3 before GBP-1 recruitment to phagosomes.

## Detection of Bla-activity by immunofluorescence microscopy

One day before infection 2.5 x 10$^4$ HeLa cells were seeded in ibidi μ-Slide 8 wells. The following day cells were infected with different bacterial strains as indicated and after 30 min the medium was replaced by 200 μl CCF4/AM loading solution (prepared according to the manufacturer's instructions) diluted in DMEM supplemented with 10% FCS and 2.5 mM probenecid. The cells were placed in the prewarmed chamber supplied with 5% CO2 of the laser scanning microscope Leica TCS SP8 and imaging was performed using a 20x oil immersion objective (NA 0.75) and the Leica LAS X SP8 software (Leica Microsystems, Wetzlar, Germany).

## Supporting information

**S1 Fig. NbALFA does not inhibit WA-314 YopD-ALFA effector translocation or cytotoxicity. (A) Digitonin assay.** Hela cells were infected with WA-314 YopD-ALFA at an MOI of 100. NbALFA-580 was diluted in cell culture medium as indicated. Cells were lyzed with digitonin and resulting supernatants (containing membrane integrated and soluble Yops from host cells) and cell pellets (containing intact bacteria and digitonin insoluble cell components) were analyzed with Western blot for the indicated proteins. YopH serves as marker for effector translocation, myc indicates myc-Rac1QL61 expression, calnexin serves as host cell loading control and HSP60 serves as bacterial loading and lysis control. Data are representative of 2 independent experiments. **(B) Cytotoxicity assay.** HeLa cells were infected for 1 h with WA-314 YopD-ALFA at an MOI of 100 and imaged by phase contrast microscopy. NbALFA was diluted in cell culture medium as indicated. Depicted are phase contrast images of a representative experiment. Scale bar: 20 μm.
(TIF)

**S2 Fig. Rac1Q61L expression enhances translocon formation by WA-314 YopD-ALFA in Hela cells.** Control or myc-Rac1Q61L transfected Hela cells were infected with WA-314 YopD-ALFA at an MOI of 50 for 50 min, fixed and stained with NbALFA-647 (shown in red) with prior permeabilization of host cell membranes using digitonin. Cells are visualized using phalloidin 488 (shown in green). Scale bar: 20 μm.
(TIF)

**S3 Fig. Live imaging of fixed and stained YopD-ALFA.** Hela cells were infected at an MOI of 50 for 50 minutes, fixed, permeabilized with digitonin and incubated with NbALFA-580 and

phalloidin 488. After addition of cell culture medium, cells were imaged with a spinning disk microscope recording z-stacks every minute for 105 min employing the same imaging conditions as for live cell imaging (e.g. for Fig 4E). The z-stacks for each time point were combined to one image using maximum intensity projection and one image every 15 min is representatively shown. The left panel shows the overview image at 0 min. The boxed region in the overview image shows the area of the video depicted in still frames to the right. Scale bars: 20 μm (overview) and 5 μm (still frames). The relative fluorescence intensity of the NbALFA-580 signals were plotted. For comparison, the live cell imaging data from Fig 4E were included in the graph.
(TIF)

**S4 Fig. Low translocon associated YopD-ALFA/YopD fluorescence signals in GFP-galectin-3 positive compartments. (A) NbALFA staining of translocon associated YopD-ALFA in fixed cells.** HeLa cells expressing myc-Rac1Q61L and GFP-galectin-3 were infected with WA-314-YopD-ALFA at an MOI of 50 for 50 min. Cells were fixed, permeabilized with digitonin and stained with NbALFA-647. Scale bars: Overview 10 μm, zoom 2 μm. (**B) Semiquantitative categorization of NbALFA signal intensities in galectin-3 positive and negative compartments.** Experimental conditions as in A. Bacteria in galectin-3 negative and positive compartments were categorized according to the intensity of YopD-ALFA signals (+/-: no/weak intensity; ++: medium intensity; +++: strong intensity). Total number of bacteria evaluated: n = 400 **(C) Immunostaining of translocon associated YopD in fixed cells.** Experimental conditions as in A but infection with WA-314 and staining with purified polyclonal anti-YopD antibody. Scale bars: Overview 10 μm, zoom 2 μm.
(TIF)

**S1 Movie. Live imaging of translocons during HeLa cell infection.** HeLa cells expressing myc-Rac1Q61L and GFP-LifeAct were infected with WA-314 YopD-ALFA at an MOI of 20 and incubated with NbALFA-580 diluted in the cell culture medium. Cells were imaged with a spinning disk microscope recording z-stacks every minute. Stacks for each time point were combined to one image using maximum intensity projection. Scale bar: 2 μm.
(AVI)

**S2 Movie. Engulfment in PIP2-positive membranes precedes translocon formation in HeLa cells.** HeLa cells expressing myc-Rac1Q61L and PLCδ1-PH-GFP were infected with WA-314 YopD-ALFA at an MOI of 20 and incubated with NbALFA-580 diluted in cell culture medium. Cells were imaged with a spinning disk microscope recording z-stacks every 20 s. Stacks for each time point were combined to one image using maximum intensity projection. Scale bar: 2 μm.
(AVI)

**S3 Movie. Engulfment in PIP2-positive membranes precedes translocon formation in primary human macrophages.** Primary human macrophages expressing PLCδ1-PH-GFP were infected with WA-314 YopD-ALFA at an MOI of 20 and incubated with NbALFA-580 diluted in cell culture medium. Cells were imaged with a spinning disk microscope recording z-stacks every minute. Stacks for each time point were combined to one image using maximum intensity projection. Scale bar: 2 μm.
(AVI)

**S4 Movie. Galectin-3 recruitment to vacuoles containing translocon forming bacteria.** HeLa cells expressing myc-Rac1Q61L and GFP-galectin-3 were infected with WA-314 Yop-D-ALFA at an MOI of 20 and incubated with NbALFA-580 diluted in cell culture medium.

Cells were imaged with a spinning disk microscope recording z-stacks every minute. The z-stacks for each time point were combined to one image using maximum intensity projection. Scale bar: 2 μm.
(AVI)

**S5 Movie. GBP-1 recruitment to galectin-3 positive bacteria.** HeLa cells expressing myc-Rac1Q61L, mScarlet-galectin-3 (shown in green) and GFP-GBP-1 (shown in red) were infected with WA-314 YopD-ALFA at an MOI of 20 and incubated with NbALFA-580 (shown in magenta) diluted in cell culture medium. Cells were imaged with a spinning disk microscope recording z-stacks every 5 minutes. The z-stacks for each time point were combined to one image using maximum intensity projection. Scale bar: 5 μm.
(AVI)

## Acknowledgments

We thank the UKE microscopy imaging facility (umif) for training and support. We thank Tomas Edgren for helpful discussion and knowing where to tag YopD.

## Author Contributions

**Conceptualization:** Martin Aepfelbacher, Manuel Wolters.

**Funding acquisition:** Martin Aepfelbacher.

**Investigation:** Maren Rudolph, Alexander Carsten, Susanne Kulnik, Manuel Wolters.

**Methodology:** Maren Rudolph, Alexander Carsten, Manuel Wolters.

**Project administration:** Martin Aepfelbacher.

**Supervision:** Martin Aepfelbacher, Manuel Wolters.

**Visualization:** Maren Rudolph, Alexander Carsten, Susanne Kulnik.

**Writing – original draft:** Maren Rudolph, Alexander Carsten, Manuel Wolters.

**Writing – review & editing:** Maren Rudolph, Martin Aepfelbacher, Manuel Wolters.

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
