## [Decision Letter · Decision Letter 0]

14 Feb 2022

Dear Dr. Wolters,

Thank you very much for submitting your manuscript "Live imaging of Yersinia translocon formation and immune recognition in host cells" for consideration at PLOS Pathogens. As with all papers reviewed by the journal, your manuscript was reviewed by members of the editorial board and by several independent reviewers. In light of the reviews (below this email), we would like to invite the resubmission of a significantly-revised version that takes into account the reviewers' comments.

The manuscript implements a novel methodology to study the dynamics of the Yersinia T3SS translocon complex. Reviewers were unanimously enthusiastic about the study. However, all 3 reviewers raised several concerns regarding technical aspects of the paper that will need to be resolved for a revised MS to be accepted. Especially noteworthy concerns raised by the reviewers center around the questions of whether ALFA tagging/ binding of the nanobody could block translocon function, whether NbALFA loss of signal is definitively reflecting loss of YopD, and why bacteria are found intracellularly in spite of their anti-phagocytic effectors. The study also implicated a role for GBPs and/or galectin-3 in YopD degradation. While these were potentially exciting preliminary findings, the reviewers criticized that the paper didn't provide direct evidence for such a model, e.g. by using cells deficient for GBPs/ Galectin-3. Exploring the biology of YopD 'degradation' further and especially probing for the functional roles of GBPs/ galectins in this process - while certainly desirable - may be beyond the scope of this study and therefore in lieu of more data, the authors could also decide to taper down the biology aspect of their paper, and revise the paper carefully to avoid making unsupported claims.

We cannot make any decision about publication until we have seen the revised manuscript and your response to the reviewers' comments. Your revised manuscript is also likely to be sent to reviewers for further evaluation.

Sincerely,

Jorn Coers

Pearls Editor

PLOS Pathogens

Raphael Valdivia

Section Editor

PLOS Pathogens

Kasturi Haldar

Editor-in-Chief

PLOS Pathogens

orcid.org/0000-0001-5065-158X

Michael Malim

Editor-in-Chief

PLOS Pathogens

orcid.org/0000-0002-7699-2064

The manuscript implements a novel methodology to study the dynamics of the Yersinia T3SS translocon complex. Reviewers were unanimously enthusiastic about the study. However, all 3 reviewers also raised several concerns regarding technical aspects of the paper that will need to be resolved for a revised MS to be accepted. Especially noteworthy concerns raised by the reviewers center around the questions of whether ALFA tagging/ binding of the nanobody could block translocon function, whether NbALFA loss of signal is definitively reflecting loss of YopD, and why bacteria are found intracellularly in spite of their anti-phagocytic effectors. The study also implicated a role for GBPs and/or galectin-3 in YopD degradation. While these were potentially exciting preliminary findings, the reviewers criticized that the paper didn't provide direct evidence for such a model, e.g. by using cells deficient for GBPs/ Galectin-3. Exploring the biology of YopD 'degradation' further and especially probing for the functional roles of GBPs/ galectins in this process - while certainly desirable - may be beyond the scope of this study and therefore in lieu of more data, the authors could also decide to taper down the biology aspect of their paper, and revise the paper carefully to avoid making unsupported claims.

Reviewer's Responses to Questions

**Part I - Summary**

Reviewer #1: The manuscript “Live imaging of Yersinia translocon formation and immune recognition in host cells” establishes a new technique using fluorescent nanobodies that recognize an ALFA tag to image the dynamics of translocon complex formation and “degradation” in situ using time lapse fluorescence imaging. The translocon complex is a major constituent of the type 3 secretion system, a molecular device acting as a molecular syringe used by many gram-negative bacterial pathogens. It is crucial for the delivery of bacterial effector proteins into the host cell. Despite its importance the translocon complex cannot easily studied, most likely due to its transient insertion into host cell membranes, and limitations working with its purified components in vitro. Therefore, the approach developed in this manuscript is welcome and has the potential to provide not only important insights into Yersinia cell invasion, however it could also be developed for other bacteria. Overall, the manuscript is written very clearly, and the presented data is well described. Together, it’s a great piece that will be very relevant for infection biologists working on secretion machines, such as the T3SS. Nevertheless, there are some overinterpretations in the text (the authors provide often only correlative data, which does not allow causal interpretations)- these issues can easily be amended down-toning some of the biological interpretations. Furthermore, the manuscript could be strengthened by some additional controls that the authors may already have, but not included in the piece, or that could be executed in a reasonable amount of time. I feel that a revised manuscript would become a very strong paper for PLoS Pathogens. Below, I state the points that should be addressed by the authors:

Reviewer #2: This manuscript takes a novel and creative approach to studying the mechanisms of Yersinia type III secretion in cells, by tagging a key translocon protein, YopD, with an ALFA tag that can be detected with nano-bodies. using this tag, the authors find that the translocon is formed in a pre-phagosomal compartment, which is not entirely surprising given that Yersinia is an extracellular pathogens and transolcates effectors from the outside of the cell. The authors furthermore use the ALFA-tagged YopD to follow the fate of translocons over time, and find that galectin-3 and GBP-1 are recruited to the phhagosome, followed by loss of the signal, suggesting degradation of the translocon. This is an intriguing approach to studying hte dynamics of translocation with a protein that is essential for the functioning of the translocon, and will likey be applicable to many other such approaches. That being said, I have some questions about the general applicability of these findings to Yersinia biology, given that the function of the Yops is to prevent phagocytosis, and that Yersinia replicates extracellularly.

Reviewer #3: In the manuscript “Live imaging of Yersinia translocon formation and immune recognition in host cells,” Rudolph et al. use live imaging studies leveraging an ALFA-tag-binding nanobodies to track the timing and localization of Yersinia enterocolitica’s YopD as a proxy for translocon formation. They demonstrate that YopD is integrated into translocons just minutes after interaction with host cells. They go on to show that YopD is associated with phosphatidylinositol-4.5-bisphosphate enriched membranes early during infection. At later timepoints they report that YopD signal colocalizes with galectin-3 and guanylate-binding protein-1 (GBP-1), which coincides with a decrease in YopD signal. The authors suggest that this might be due to degradation of the translocon via galectin-3 and/or GBP-1 mediated pathway.

The questions tackled by these authors is important: understanding the kinetics and location of translocon formation will provide insights into Yersinia virulence mechanisms. Their live imaging approach is particularly exciting, as it may be adaptable to the study of other bacterial pathogens that encode TTSSs/translocons. Their report that translocon colocalization with Gal-3 and GBP1 coincides with loss of YopD signal, is an intriguing finding. That said, there are several important caveats that should be addressed to fully support the conclusions drawn in this manuscript.

**Part II – Major Issues: Key Experiments Required for Acceptance**

Reviewer #1: I would suggest experimental data on the two points below:

-Figure 1 provides convincing controls to prove that insertion of the ALFA tag does not perturb the translocon activity. However, the rest of the paper focuses on imaging of the translocon insertion, requiring the binding of a nanobody (NbALFA) to the ALFA tag. Therefore, the authors should add some data that the binding of the nanobody does not entirely block translocon function. This could be included in Figure 1.E with an experiment in the presence of the NbALFA. Similar comments could be made for Figure 1.F.

- Figure 3 onwards (important to be addressed carefully): The present study relies on observation/quantification of gain and loss of signals through the nanobody binding to the ALFA tag. The authors state that using different intervals of imaging with similar results of their data indicate limited or no photobleaching. This cannot be interpreted as such, as photobleaching relies on different parameters, for example the intensity of the incoming light (instead of the sequence). As this is such an important issue in the manuscript, additional controls should be performed. For example, this could be done by co-straining in an indirect immunofluorescence experiment with NbALFA and an anti YopD antibody at early and again at late infection times after the end of their time-lapse. Failure to detect an antibody signal at later infection stages will provide additional evidence that the NbALFA loss of signal is not due to photobleaching but really to YopD disappearance.

Reviewer #2: 1. Injection of Yops E and H blocks phagocytosis. If the translocon is functioning appropriately, how is Yersinia entering the cell? I recognize that some portion of Yersinia are indeed inside the cell, but these would presumably be bacteria whose T3SS was not functioning as well. It is important to know how the conclusions here impact the extracellular bacteria that are presumably the pool of replicating ones.

2. The authors conclude from the loss of YopD fluorescence signal that it is being degraded. Is it fair to conclude that? Could recruitment of galectin and GBPs to the phagosome and bacterial membrane interfere with the ability of the nanobody to bind to YopD? Is there a way to measure the levels of YopD by western blotting of digitonin-treated cells which should provide access to the extracellular YopD pool?

3. Can the authors perform these studies in the absence of Gbps (GbgpChr3-/- cells for example?) a prediction of their model is that the translocon should not be degraded in these cells.

4. Given that YopD itself is translocated, how do the authors know that the YopD they are imaging is part of a functional translocon, rather than YopD that is being translocated into the cytosol of the target cell? This seems important to address to be able to support the conclusion being reached. If the translocon is degraded in a GBP-dependent manner, presumably in the absence of GBPs, that degradation would not occur, and there should be more translocated effectors, which could be measured by beta-lactamase assay.

Reviewer #3: Overall the manuscript is written well, however there is important details that are lacking throughout the manuscript that make the experimental design/data/figures hard to understand. For example: there was limited information about the strain background. Was the tagged YopD complementing a deltaYopD mutant or was there an endogenous copy? I also had to dig for the MOIs, conditions for the secretion assay, beta-lactamase reporter, etc. More detail in the results section is absolutely required to enhance the manuscript.

The major advance of this manuscript is the ability to potentially visualize the formation, localization and overall dynamics of the translocon. Since this is a new technique that the entire manuscript hinges on, there needs to be additional evidence to indicate that the authors are indeed visualizing the translocon. One way this could be addressed is via biochemical isolation of phagosomal membranes, followed by western blot analysis probing for YopD at key time points. This type of experiment would support their conclusion that they are indeed visualizing YopD/the translocon (enrichment of the protein in the membrane fraction should coincide with visualization of ALFA-tag-binding nanobodies in their time-lapse movies). In addition, further modulating of the system would help support the conclusion that they are the visualizing YopD/the translocon. For example, the authors state that cells expressing myc-Rac1Q61L strongly increases translocon expression. Does expression of myc-Rac1Q61L (compared to controls) increase YopD signal in the live imaging system?

The authors find YopD signal colocalizes with galectin-3 and guanylate-binding (GBP-1), which coincides with a decrease in YopD signal. The authors suggest that this might be due to degradation of the translocon via galectin-3 and/or GBP-1 mediated pathway. This is potentially very interesting and the dependency is testable. It would greatly enhance their findings if the authors can conclude that YopD half-life is impacted by galectin-3 and GBP-1. Using gal-3 and GBP-1 knockdowns or knockouts and visualizing the spaciotemporal localization of YopD via live cell imaging/western blot would help draw such conclusions. In addition, modulating the system (e.g. inhibiting vacuole acidification or macrophage activation) and measuring how YopD signal changes over time would give additional support. This set of experiments will help give deeper insights into the relationship of Gal-3 and GBP-1 to YopD. Without this data this is simply an observation that the Yersinia translocon colocalizes with markers of membrane damage.

**Part III – Minor Issues: Editorial and Data Presentation Modifications**

Reviewer #1: -Figure 2.B – The STED data is very good and relates very well to previous work of the authors using translocon specific antibodies. I am wondering how to interpret the dot in the middle of the bacterium, as it appears that this translocon is not fully in focus. Therefore, the authors should include a 3D STED dataset or they should use a dataset where the translocons out of focus do not complicate the interpretation of the data.

-Figure 3. The authors state that T3SS translocon pore is inserted after PIP2 enrichment of the vacuolar membrane. I am wondering whether this recruitment does already require some T3SS effector injection. As PIP2 is normally strongly enriched at the plasma membrane (it is well depicted on Figure 3C), and then it gets depleted during phagosome formation, one wonders when does the T3SS really start to deliver effectors… the authors should discuss possible limitations of the fluorescence signals and also relate this to previous work from the Grinstein lab.

Figure 4G: The GBP signal does not label all phagosomes/vacuoles that contain bacteria- this is interesting and the authors may want to discuss this issue.

Movie relating to 4G and figure: at later time points, there are unspecific signals of the nanobody channel- what are these signals? They are not translocons that are carried away to some other compartments (for example for degradation?)? Please discuss this issue.

The authors use the myc-Rac1Q61L cell line which provides a boost in translocon insertion and the observed infection dynamics. It has been used by the authors before, and it would be good to explain this again in this work, so that the reader can follow why it was used.

Important general comment: The paper tends to over-interpret some observations without providing additional experiments needed to prove such statements. Of course, it would be possible to address this with extra experiments, however, as the work appears as a proof-of-concept method work, one could down-tone the writing, and suggest experiments to really proof “causal” relationships in the future. A number of examples for this are

*Lines 166-169: It is not clear whether PIP2 insertion is required for translocon insertion from the shown experiments. The data is a simple correlation. Here, experiments with inhibitors or kinases and phosphatases would be required to really mark the point. Nevertheless, this could simply be discussed.

Similar issues

*Lines 192-193: here again, two fluorescence signals are correlated, but a causal link is beyond the acquired data. This can easily be discussed.

*Lines 207-209: the shielding of GBP to avoid antibody binding is speculative and would require more controls- again, this could be suggested and elaborated in the discussion.

Reviewer #2: (No Response)

Reviewer #3: (No Response)

PLOS authors have the option to publish the peer review history of their article (what does this mean?). If published, this will include your full peer review and any attached files.

Reviewer #1: No

Reviewer #2: No

Reviewer #3: No
---

## [Editor Report · Decision Letter 1]

21 Apr 2022

Dear Dr. Wolters,

We are pleased to inform you that your manuscript 'Live imaging of Yersinia translocon formation and immune recognition in host cells' has been provisionally accepted for publication in PLOS Pathogens.

Best regards,

Jörn Coers

Pearls Editor

PLOS Pathogens

Raphael Valdivia

Section Editor

PLOS Pathogens

Kasturi Haldar

Editor-in-Chief

PLOS Pathogens

orcid.org/0000-0001-5065-158X

Michael Malim

Editor-in-Chief

PLOS Pathogens

orcid.org/0000-0002-7699-2064

The authors put in place a convincing method to study the dynamics of the insertion of the bacterial type 3 secretion system translocon into the plasma membrane of infected cells. The paper constitutes an important technical advance for the field and the reviews were overwhelmingly positive. The minor technical concerns raised by the reviewers were convincingly addressed in the revisions
---

## [Editor Report · Acceptance letter]

18 May 2022

Dear Prof. Aepfelbacher,

We are delighted to inform you that your manuscript, "Live imaging of *Yersinia* translocon formation and immune recognition in host cells," has been formally accepted for publication in PLOS Pathogens.

Best regards,

Kasturi Haldar

Editor-in-Chief

PLOS Pathogens

orcid.org/0000-0001-5065-158X

Michael Malim

Editor-in-Chief

PLOS Pathogens

orcid.org/0000-0002-7699-2064